# High-Throughput Sequencing-Based Analysis of Changes in the Vaginal Microbiome during the Disease Course of Patients with Bacterial Vaginosis: A Case–Control Study

**DOI:** 10.3390/biology11121797

**Published:** 2022-12-10

**Authors:** Jing Gao, Yiqian Peng, Nanyan Jiang, Youhao Shi, Chunmei Ying

**Affiliations:** Department of Clinical Laboratory, The Obstetrics & Gynecology Hospital of Fudan University, Shanghai 200011, China

**Keywords:** vaginal microecology, bacterial vaginosis, 16S rRNA, *Lactobacillus*, treatment, vaginal microbiome

## Abstract

**Simple Summary:**

Because of their complexity, bacterial communities characteristic of bacterial vaginoses (BV) have not been completely characterized yet. The aim of the present study was to evaluate the changes in different characteristic bacteria during the onset, progression, and remission of BV. We performed a case–control study to investigate the differences in vaginal microbiota during the onset and post-treatment asymptomatic stages of BV. Participants were divided into four groups. We collected vaginal swabs and sequenced the V3–V4 hypervariable regions of bacterial 16S rRNA genes using the Illumina MiSeq platform. The sequencing results were used to evaluate vaginal microbiomes in the four groups. We observed significant differences in the composition and alpha diversity of the vaginal microbiota at different stages of BV, as well as in the distribution of bacterial communities among the investigated groups. We commented on these results and proposed future research directions. We believe that our study makes a significant contribution to the literature because it provides a basis for the screening of possible specific markers of BV, as well as for the prevention, diagnosis, treatment, and efficient monitoring of the disease in clinical practice.

**Abstract:**

Background: The vaginal microbiome is closely associated with the onset and recurrence of bacterial vaginosis (BV). In the present study, the state of vaginal microbiota during the onset and post-treatment asymptomatic stages of BV were compared to that of a healthy population to evaluate the changes in different characteristic bacteria during the onset, progression, and remission of BV. Methods: A case–control study was performed to explore these changes. Women with clinical symptoms of BV were divided into the disease group (M) and case–control group (C) based on the Nugent score. Subjects in the disease group whose symptoms were resolved after the treatment were assigned to the treated group (T) and healthy subjects were recruited into the normal control (N) group. The V3–V4 hypervariable regions of bacterial 16S rRNA genes were sequenced on the Illumina MiSeq platform. Results: The N harbored the highest number of detected species and a higher abundance of microbiota; they had a significantly higher abundance of *Lactobacillus* and different bacterial community composition compared to the other three groups. In group M, *Gardnerella vaginalis* was the dominant species, whereas *Lactobacillus iners* was predominant in the other three groups. While *Lactobacillus* was more commonly present in Group C compared to group M. it was significantly increased in group T. Alpha diversity analysis of bacterial communities revealed significant differences in community richness and diversity among all four groups (*p* < 0.05). Significant differences in the distribution of various bacterial communities among the different groups were also observed (*p* < 0.05). Specifically, the abundance of eight bacterial taxa (*Megasphaera*, *Aerococcus christensenii*, *Clostridiales*, *Gardnerella*, *Peptostreptococcus*, *Veillonellaceae*, *Akkermansia*, *Coriobacteriales*) differed significantly among the four groups (*p* < 0.05). Conclusion: Significant differences in the composition and alpha diversity of the vaginal microbiota at different stages of BV and the distribution of bacterial communities were observed among the investigated groups. In addition to *Gardnerella*, *Sneathia sanguinegens* and *Prevotella timonensis* play an important role in the pathogenesis of BV. The appearance of BV-like clinical symptoms was closely associated with the decrease in *Prevotella* and *Atopobium vaginae* populations.

## 1. Introduction

Bacterial vaginosis (BV) is characterized by a decrease in the number of normal hydrogen peroxide-producing lactobacilli and an overgrowth of anaerobic bacteria, such as *Gardnerella vaginalis*, *Bacteroides*, and *Prevotella,* in the vagina [1]. Its manifestations include an increase in vaginal pH, an increase in vaginal discharge with a fishy odor, and mild vulvar pruritus. Currently, 75% of women worldwide have been affected by vaginitis at least once in their lives, with pH being closely related to the vaginal ecosystem [2] and BV being the most common vaginal microbiota dysbiosis affecting female reproductive health. Currently, the gold standard for diagnosis of BV in clinical laboratories is the Nugent score, which is calculated via microscopic examination of the relative numbers of *Lactobacillus*, *Gardnerella vaginalis*, *Bacteroides*, and *Mobiluncus* using Gram-stained smear specimens. The mechanisms of BV occurrence and recurrence are extremely complex. BV was first reported to be associated with *G. vaginalis* in 1955; however, the debate surrounding the etiology of BV still persists, involving extensive studies on human or animal models using *G. vaginalis* [3]. BV is closely associated with the use of antimicrobial agents, the transmission of bacterial communities by sexual partners, and the presence and eradication of BV-related biofilms [4], as well as conditions such as infertility, endometritis [5], chronic pelvic inflammatory disease, and preterm delivery [6]. Furthermore, BV is associated with an increased risk of HIV infection and other sexually transmitted diseases [7]. Patients with BV account for up to one-third of gynecology outpatients in China; the United States Food and Drug Administration (U.S. FDA) has reported figures of 40–50%, which are related to sexual activity and promiscuity. The Sexually Transmitted Infections (STI) Treatment Guidelines released by the United States Centers for Disease Control and Prevention (U.S. CDC, July 2021), the Guidelines for Diagnosis and Treatment of Bacterial Vaginosis (2021 revised edition), and the Infectious Diseases Collaborative Group of the Society of Obstetrics and Gynecology of the Chinese Medical Association recommend clinicians to use metronidazole and clindamycin for the treatment of BV.

Due to the current popularity of research on the gut–vagina axis, researchers have been actively investigating the use of new probiotic strains or lactoferrin as non-pharmacological strategies for counteracting vaginal infections [8]. These strategies also contribute to the maintenance of homeostasis in the vaginal microbiota [9]. Certain probiotic *Lactobacillus* strains can be used to prevent and treat BV and vulvovaginal candidiasis [10]. *Lactobacillus iners* plays a vital role in the vaginal microbiota; it possesses several probiotic characteristics while also seeming to be an opportunistic pathogen, the mechanisms underlying which warrant further studies [11]. Bacterial communities undoubtedly exhibit a profound influence on the vaginal microbiome, with different ethnic groups exhibiting a high variability in vaginal microbiota composition. Genomics significantly contributes to the colonization and persistence of *Lactobacillus* strains but has some limitations [12]. Although the widespread presence and significance of *Lactobacillus* have already been widely reported, research progress remains hindered by the complexity and variability of vaginal microbiota.

In the present study, we investigated the state of vaginal microbiota during the onset and post-treatment asymptomatic stages of BV in women with symptoms of BV. We further compared the data obtained to those in the normal healthy population to evaluate the changes in different characteristic bacteria during the onset, progression, and remission of BV. Our findings provide a scientific basis for the screening of possible characteristic markers of BV, which are of great significance for the prevention, diagnosis, treatment, and efficacy of monitoring BV in clinical practice.

## 2. Materials and Methods

### 2.1. Study Population

Non-pregnant women of childbearing age who had sought medical consultation at the gynecology clinic or cervical clinic of Obstetrics and Gynecology Hospital of Fudan University, China, between April 2018 and April 2019 were screened for eligibility to participate in the study. The exclusion criteria were as follows: (1) women with a history of cervical cancer or other lower genital tract malignancies; (2) women who had previously undergone hysterectomy, destructive treatment of the cervix, or other cervix treatments; (3) women with a positive test result for HIV, hepatitis B, or hepatitis C; (4) women with a concomitant autoimmune disease; (5) women who were menstruating or had engaged in sexual intercourse or vaginal douching 48 h before sample collection; (6) women with *Trichomonas vaginalis* or *Candida* infection. Patients with the following clinical symptoms of BV were included in the study: (1) increased vaginal discharge, which may be accompanied by mild pruritus or burning sensation in the vulva; (2) greyish-white, homogeneous, and thin discharge that usually adheres to the vaginal wall; (3) absence of hyperemia of the vaginal mucosa, which is a manifestation of inflammation.

This study included 40 subjects, who were divided into four groups: (1) normal control group (N) consisting of healthy subjects (*n* = 12). Healthy controls included individuals who underwent a routine physical examination at our hospital during the same time period as the study, had no history of BV or clinical symptoms of infection within 6 months, received normal results of routine examination of vaginal discharge with no *Trichomonas*, *Candida*, or clue cells detected, and received no antibiotic treatment within 2 weeks. (2) Case–control group (C) comprised subjects (*n* = 11) with clinical symptoms of BV and a Nugent score < 7. (3) Disease group (M) included subjects (*n* = 9) with clinical symptoms of BV and a Nugent score ≥ 7. (4) Treated group (T) involved subjects (*n* = 8) of the disease group whose clinical symptoms of BV had been resolved after treatment with metronidazole or clindamycin, with no BV recurrence within 6 months. The characteristics of the four groups are presented in Table 1.

### 2.2. Sample Collection

A sterile, disposable speculum was inserted into the vagina without lubricant. Subsequently, a swab was collected from the posterior vaginal fornix and stored in a sterile empty 3mLEP tube, which was immediately placed on ice and transferred to the laboratory, where it was stored at −80 °C.

### 2.3. DNA Extraction and PCR Amplification

Microbial DNA was extracted from vaginal swabs using the E.Z.N.A.^®^ Soil DNA Kit (Omega Bio-tek, Norcross, GA, USA), according to the manufacturer’s protocol. Final DNA concentration and purification were determined using a Nano Drop 2000 UV-vis spectrophotometer (Thermo Scientific, Wilmington, DE, USA), and DNA quality was verified using 1% agarose gel electrophoresis. The V3-V4 hypervariable regions of the bacterial 16S rRNA gene were amplified with the primers 338F (5′-ACTCCTACGGGAGGCAGCAG-3′) and 806R (5′-GGACTACHVGGGTWTCTAAT-3′) using a thermocycler PCR system (GeneAmp 9700, ABI USA, Los Angeles, CA, USA). The PCR reactions were performed using the following program: denaturation at 95 °C for 3 min, 27 cycles of 30 s at 95 °C, annealing at 55 °C for 30 s, elongation at 72 °C for 45 s, and a final extension at 72 °C for 10 min. The PCR reactions were performed in triplicate using a 20 μL mixture containing 4 μL of 5× FastPfu buffer, 2 μL of 2.5 mM dNTPs, 0.8 μL of each primer (5 μM), 0.4 μL of FastPfu polymerase, and 10 ng of template DNA. The obtained PCR products were extracted from a 2% agarose gel, subsequently purified using the AxyPrep DNA Gel Extraction Kit (Axygen Biosciences, Union City, CA, USA), and quantified using QuantiFluorTM -ST (Promega, Madison, WI, USA).

### 2.4. Illumina MiSeq Sequencing

The purified amplicons were pooled in equimolar concentrations and paired-end sequenced using the Illumina MiSeq platform (Illumina, San Diego, CA, US), according to the standard protocols by Majorbio Bio-Pharm Technology Co. Ltd. (Shanghai, China).

### 2.5. Processing of Sequencing Data

The raw fastq files were quality-filtered by Trimmomatic and merged using FLASH with the following criteria: (i) the reads were truncated at any site receiving an average quality score < 20 over a 50 bp sliding window; (ii) the sequences whose overlap was longer than 10 bp were merged according to their overlap with a mismatch of no more than 2 bp; (iii) the sequences of each sample were separated according to barcodes (exact matching) and primers (allowing two nucleotide mismatch), and the reads containing ambiguous bases were removed. Operational taxonomic units (OTUs) were clustered with a 97% similarity cutoff using UPARSE (version 7.1; http://drive5.com/uparse/ (accessed on 7 June 2019)), which has a novel greedy algorithm that performs chimera filtering and OTU clustering simultaneously. The taxonomy of each 16S rRNA gene sequence was analyzed using an RDP classifier algorithm (http://rdp.cme.msu.edu/ (accessed on 7 June 2019)) against the Silva 16S rRNA database (Release128, http://www.arb-silva.de (accessed on 7 June 2019)) using a confidence threshold of 70%.

### 2.6. Statistical Analysis

Alpha, Chao, and Shannon’s diversity indices were calculated. Wilcoxon rank sum test was adopted for comparison between groups, Kruskal–Wallis rank sum test was used to identify features with a significantly different abundance of taxa among the groups, and Pearson’s chi-squared test was used to compare the vaginal community state types (CSTs) of the four groups. Differences were considered statistically significant when the *p*-value was less than 0.05.

## 3. Results

### 3.1. Characteristics of the Study Population

Among the four groups, the differences in age (*p* = 0.260), human papillomavirus (HPV) infection status (*p* = 0.408), and pathological examination results (*p* = 0.271) were not statistically significant, whereas the differences in the Nugent score and *Lactobacillus*, *G. vaginalis*, and *Bacteroides* detection were significant (*p* < 0.001). Microscopic examinations of the stained smear samples indicated the absence of *Mobiluncus* species in all four groups (Table 1).

### 3.2. Sequencing Results

Samples from all 40 subjects were subjected to diversity analysis. A total of 1,940,658 optimized sequences with 814,893,068 bases (bp) and an average sequence length of 419.905551622 bp were obtained, and 1674 operational taxonomic units were detected (Table 2).

### 3.3. Species Composition

The bacterial species composition of the samples obtained from the four groups of patients was analyzed. A total of 1511, 422, 209, and 316 species were detected in samples collected from the N, C, M, and T groups, respectively (Figure 1).

### 3.4. Community Composition

The bacterial community composition analysis revealed significant differences among the samples from the four groups. *L. iners* was the dominant bacterium in N, C, and T groups, whereas *G. vaginalis* was the dominant species in the M group. The number of *Lactobacillus jensenii* and *Lactobacillus gasseri* was significantly higher in the C group than that in the M counterpart. A significant increase in the number of *Lactobacillus* was observed in the T group after pharmacological treatment. The N group exhibited a significantly higher number of *Lactobacillus* and significantly different community composition, with slightly higher numbers for *Megasphaera*, *Aerococcus christensenii*, *Prevotella*, *Clostridiales*, *Sneathia sanguinegens*, *Atopobium*, *Enterobacteriaceae*, and *Streptococcus* than those in the other three groups (Figure 2).

### 3.5. Vaginal Microbiome Richness and Diversity

Alpha diversity of the bacterial communities of the four groups was analyzed using Student’s *t*-test on estimators of species richness, coverage, and diversity within the communities. Differences in sequencing depth among the groups were not statistically significant (*p* > 0.05). Bacterial community richness in the N group differed significantly from that in the other three groups (*p* < 0.05); furthermore, community diversity of the C group differed significantly from that in the M and N groups (*p* < 0.005), and community diversity in the M group differed significantly from that in the T group (*p* < 0.05; Table 3 and Figure 3).

### 3.6. Typing Analysis at Phylum Level

Typing analysis of the bacterial communities for the different groups was performed at the phylum level. Intragroup community compositions were relatively consistent, with most samples lying within the confidence interval (Figure 4). In contrast, intergroup differences in community composition were statistically different.

### 3.7. Differences in Bacterial Communities among Different Groups

The distribution of bacterial communities was significantly different among the different groups. *G. vaginalis* was relatively more abundant in the M and T groups (*p* < 0.005); *Lactobacillus* was hardly detected in the M and T groups but significantly abundant in the C group (*p* < 0.01); *Prevotella* was hardly detected in the C group (*p* < 0.01); lower abundance of *Atopobium vaginae* was noted in the C group than that in the other groups (*p* < 0.05); *S. sanguinegens* and *p. timonensis* were more abundant in the M group than in the other groups; and the abundance of *Megasphaera*, *A. christensenii*, *Clostridiales*, *Gardnerella*, *Peptostreptococcus*, *Veillonellaceae*, *Akkermansia*, *Coriobacteriales*, and *Veillonellaceae* were significantly different among the four groups (*p* < 0.05; Figure 5). In addition to *Gardnerella*, *S. sanguinegens* and *p. timonensis* play an important role in the pathogenesis of BV. Decreased levels of *Prevotella* and *A. vaginae* were closely associated with the appearance of BV-like clinical symptoms.

## 4. Discussion

The unique microbial community in the female lower genital tract is commonly known as the vaginal microbiota. It exhibits significant geographical differences in composition and richness, serves an important role in reproductive cyclicity, provides significant benefits to the prevention of preterm delivery and infections, and also improves the efficacy of treatments for vaginal tumors [13]. In bacterial vaginosis (BV), the characteristics of vaginal microbiota are highly complex. Next-generation sequencing techniques have been widely used for the detection of pathogenic microbes [14]. In the present study, we used 16S rRNA gene sequencing to investigate the vaginal microbiome of patients with BV. We adopted a case–control approach to determine the response of patients with BV to pharmacological treatment by comparing the differences in vaginal microbiota between healthy subjects and patients whose clinical symptoms of BV were significantly alleviated after treatment with metronidazole and clindamycin. The results of our study are of great significance for the diagnosis, treatment, and prognosis of BV.

*Lactobacillus*-dominated vaginal microbiota have been previously regarded as beneficial to vaginal health because an imbalance of the vaginal microenvironment is usually caused by a lack of lactobacilli and an overgrowth of anaerobic bacteria. However, an excessive number of lactobacilli is associated with cytolytic vaginosis, and the state of various microbes during BV progression affects the clinical presentation and pathogenesis of the disease. Furthermore, the heterogeneity and diversity within the genus *Gardnerella* may impact BV progression [15].

In the current study, samples from non-pregnant women of childbearing age who had sought medical consultation at a gynecology clinic were collected. Patients with factors that may affect the vaginal microbiota, including a history of cervical cancer or other lower genital tract malignancies, were excluded from the study. Cytokines, such as TNF-α and MIP-1β, are associated with the progression of BV to cervical intraepithelial neoplasia (CIN) [16]. The effects of infection by common pathogenic microbes (e.g., vaginal HPV) and pathological classification were also excluded. At present, laboratory-based diagnosis of BV performed in clinical, microbiological laboratories is largely reliant on the Nugent scoring method, which is regarded as the gold standard for BV detection [17]. Mykhaylo Usyk et al. recently reported the development of a 16S rRNA gene amplicon sequencing-based algorithm named *molBV* for BV diagnosis [16]. New diagnostic methods, including highly sensitive and specific point-of-care (POC) tests, have also been developed based on the identification of biomarkers from vaginal microbiome and vaginal metabolome data [18]. Our results indicated that the Nugent scores for the participating subjects were consistent with those obtained from 16S rRNA gene sequencing, which further confirmed the accuracy of Nugent scoring in BV diagnosis. Furthermore, our data revealed that the N group had significantly higher species diversity than the other groups. The M group had the lowest number of detected species, which significantly improved in treated subjects in the T group compared with that in the untreated subjects. Although the subjects in the C group also exhibited clinical symptoms of vaginal dysbiosis, the state of the vaginal microbiota was slightly better than that in the M group.

Furthermore, significant differences were noticed in the community composition of all four groups. The dominant bacterium in the M group was *G. vaginalis*, which was in accordance with the characteristics of laboratory-based diagnosis of BV. *G. vaginalis* effectively displaces lactobacilli and adhere to vaginal epithelial cells and exhibits an increased propensity for biofilm formation, which increases the chances of infection. However, recent data suggested that *G. vaginalis* may be necessary but not sufficient for BV development because colonization with this species does not always lead to BV [19]. Thirteen species of the genus *Gardnerella* have been recognized to date, but whether BV is caused by the main pathogen, *G. vaginalis,* or a microbiota comprising multiple sexually transmitted microbes remains under debate [20]. Researchers have recently proposed a new conceptual model hypothesizing that *G. vaginalis*, *Prevotella bivia*, and *A. vaginae* are jointly involved in BV pathogenesis. The current study showed that *L. iners* was the dominant species in all groups except for the M group. Besides its role as a probiotic, *L. iners* may also be an extremely important opportunistic pathogen [11]; its functions have always been the subject of much debate. Our results indicated that the numbers of *L. jensenii* and *L. gasseri* in the C group were significantly higher than those in the M counterpart. This suggested that the bacterial community structures of the C and M groups were different despite the similarities in clinical symptoms between the two groups. Therefore, confirmation by laboratory testing is required prior to the diagnosis of BV by clinicians to avoid the blind use of antibacterial agents based on experience, which will delay disease treatment. Moreover, the pharmacological treatment significantly increased the abundance of *Lactobacillus*. In the N group, the number of *Lactobacillus* was significantly higher than that in the other three groups, among which the bacterial community composition was also significantly different. This emphasized the importance of *Lactobacillus* in the maintenance of vaginal homeostasis. Compared to the other three groups, the N group also had slightly higher numbers of *Megasphaera*, *A. christensenii*, *Prevotella*, *Clostridiales*, *S. sanguinegens*, *Atopobium*, *Enterobacteriaceae*, and *Streptococcus*, as well as significantly higher community richness; moreover, there were significant differences in community diversity of the N group and that of the C and M groups. These results suggested that the diversity and richness of bacterial communities play key roles in the maintenance of vaginal health. Differences in community diversity between the M and T groups, as revealed by the analysis of alpha diversity indices, also provided further validation of the effectiveness of the classic metronidazole and clindamycin regimens in treating BV.

Previous research has indicated that the roles of certain rare bacteria or non-readily cultivable bacterial strains in maintaining vaginal homeostasis are usually underestimated. The widespread application of molecular biology techniques, especially high-throughput sequencing, in the field of pathogenic microbe detection has provided excellent tools for the in-depth investigation of the roles of various microbes in vaginal homeostasis. Furthermore, it significantly impacts the evaluation of every microbe and the relationships among their different physiological stages, for instance, during pregnancy [21]. Bioinformatics analyses facilitate the use of these techniques in microbe identification and functional research, which are lacking in traditional culturing methods. Our results showed that the relative abundance of *G. vaginalis* in the vaginal microbiota was higher during the onset of BV and after treatment (*p* < 0.005), whereas *Lactobacillus* was hardly detected. However, we found that the abundance of *Lactobacillus* was higher in the vaginal microbiota of patients with BV-like symptoms (*p* < 0.01) than in that of the other patients. The abundance of *A. vaginae* was lower in the C group (*p* < 0.05) than in the other three groups; the abundance of eight types of bacteria (*Megasphaera*, *A. christensenii*, *Clostridiales*, *Gardnerella*, *Peptostreptococcus*, *Veillonellaceae*, *Akkermansia*, and *Coriobacteriales*) differed significantly among the four groups.

Besides focusing on the major bacterial communities of the vaginal microbiota, adequate attention should also be paid to certain rare bacteria and emerging pathogens detected in the present study. In the past, the detection of *Sneathia* by traditional culturing methods had been immensely difficult because of its complex nutritional requirements, extremely slow growth, and exceptionally stringent growth conditions. This has led to the scarcity of studies on this genus. However, the use of molecular biology methods in recent years has spurred progress in research related to *Sneathia*. A case–control study indicated that *Sneathia* was closely associated with recurrent spontaneous abortion (RSA), and metformin combined with aspirin for the treatment of RSA significantly increased the relative abundance of vaginal *Lactobacillus* spp. [22]. Additionally, *Sneathia* contributes to the pathogenesis of BV, chorioamnionitis, preterm prelabor rupture of the membranes, spontaneous preterm labor, stillbirth, maternal and neonatal sepsis, HIV infection, and cervical cancer [23]. Interestingly, in the present study, the abundance of *S. sanguinegens* was relatively higher in the C group than that in the other groups, indicating the presence of significant differences between BV and diseases with BV-like clinical symptoms. Therefore, the roles of *Sneathia* species in the vaginal microbiota warrant further investigation.

Furthermore, *G. vaginalis* and *Prevotella*, which are early colonizers in BV, were hardly detected in the N group. *Prevotella* species induce an immune response and the release of inflammatory mediators from various stromal cells, which are associated with increased augmented T helper type 17 (Th17)-mediated mucosal inflammation [24]. The roles of *A. vaginae* and other BV-associated bacteria as secondary colonizers in BV also require further investigation [19]. Our results indicated that the independent or interactive effects of these bacteria are directly related to vaginal microbiota dysbiosis, but further research is required to elucidate their relationships with BV.

A great concern to clinicians in BV treatment is the high post-treatment recurrence rate. Currently, no published data directly assess the role of non-adherence in poor treatment outcomes and recurrent BV [25]. Concurrent male partner treatment can significantly improve the microbiota in the female vagina as well as that in the male penile skin and urethral orifice [26], thereby achieving improved BV treatment outcomes [27]. Zhang et al. studied the use of probiotics in BV treatment and reported that the oral administration of *Lacticaseibacillus rhamnosus* GR-1 and *Limosilactobacillus reuteri* RC-14 for 30 days as an adjunct therapy in BV did not enhance the disease cure rate [28]. However, stabilizing the low diversity healthy flora by promoting the growth of health-associated *Lactobacillus* spp., such as *Lactobacillus crispatus,* may be beneficial for long-term female health [29]. Increased specificity of *L. crispatus* is strongly associated with the clearance of high-risk HPVs [30]. Happel et al. showed that vaginal *Lactobacillus* strains performed better in in vitro assays than the probiotic strains currently used in probiotics for vaginal health, and their effective use may result in improved BV treatment options. Including the best-performing vaginal *Lactobacillus* isolates in a region-specific probiotic for vaginal health might result in improved BV treatment options, and this has been proved by the relevant research on vaginal *Lactobacillus* isolates from South African women [31]. Cohen et al. reported that the use of *L. crispatus* CTV-05 (Lactin-V) after treatment with vaginal metronidazole resulted in a significantly lower incidence of BV recurrence after 12 weeks [32]. Another study explored the use of a vaginal pessary containing glucono delta-lactone (GDL) and sodium gluconate (NaG) as an adjuvant therapy for BV and found that it contributed to the restoration of normal pH and disruption of the associated biofilm [33]. We believe that *Lactobacillus* provides immense benefits to disease treatment and prognosis and expect further research to progress into this area in the future.

This study had certain limitations: (1) Only a small number of patients were included in the four groups due to the difficulties in case matching and follow-up after treatment. (2) The effects of the bacterial communities of sexual partners on the vaginal microbiota of the included subjects were not considered. (3) Besides using high-throughput sequencing techniques, we also excluded interfering factors to the greatest extent possible when defining the subject inclusion and exclusion criteria; however, certain related indicators, such as inflammatory cytokines and patient immune status, were not measured in this study. (4) When we selected patients for the group, immunochromatography was used to screen for *Chlamydia trachomatis,* and the negative subjects were selected. However, we found that the sensitivity of immunochromatography was low after detecting the organism using the sequencing method; therefore, asymptomatic carriers could not be identified. This can be prevented by employing the fluorogenic quantitative PCR method for pathogen screening. Despite these limitations, we believe that the results of this study can serve as a key reference for clinicians to understand the state of the vaginal microbiota during the onset and progression of BV and provide a basis for future research on the screening of characteristic markers of BV.

## 5. Conclusions

The diversity and richness of bacterial communities are crucial in the maintenance of vaginal health. Significant differences were observed in the composition and alpha diversity of the vaginal microbiota at different stages of BV and the distribution of bacterial communities among the four groups.

## Figures and Tables

**Figure 1 biology-11-01797-f001:**
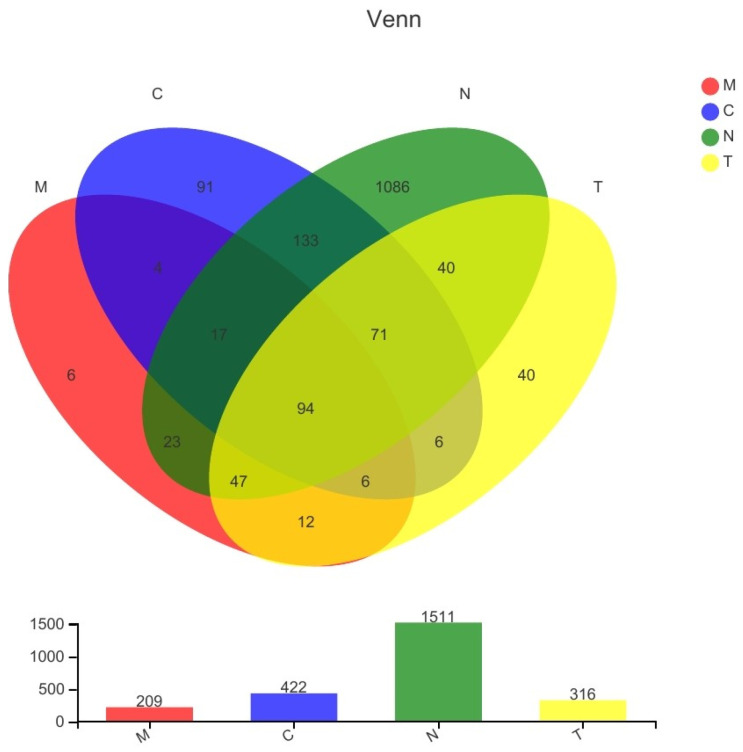
Venn diagram of bacterial species detected in samples collected from all four groups.

**Figure 2 biology-11-01797-f002:**
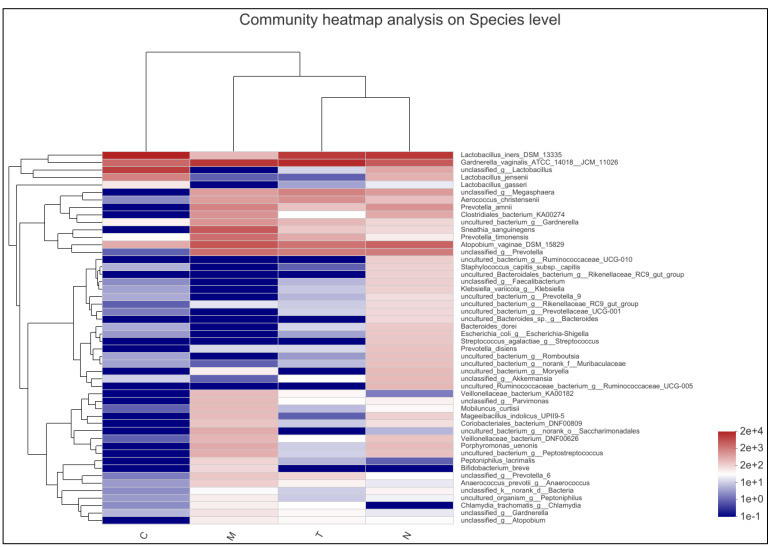
Results of community heatmap analysis at species level, with the group names and species names plotted on the *x*-axis and *y*-axis, respectively. Color gradients indicate the changes in the abundance of various samples across the different groups, and the values represented by the color gradients are shown on the right side of the heatmap. 2e+4 = 2 × 10^4^; 2e+3=2 × 10^3^; 2e+2 = 2 × 10^2^; 1e+1 = 1 × 10^1^; 1e+0 = 1 × 10^0^; 1e−1 = 1 × 10^−1^.

**Figure 3 biology-11-01797-f003:**
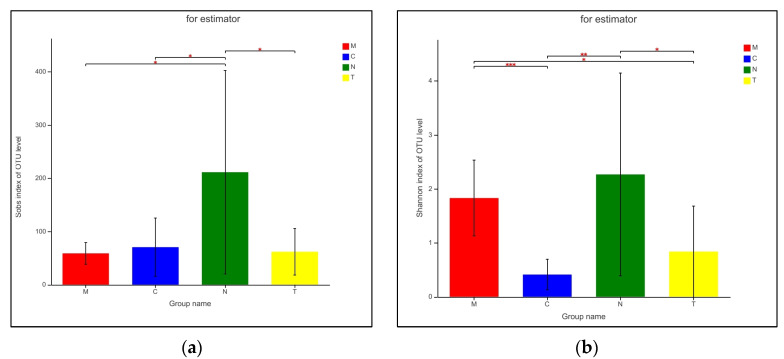
Vaginal microbiota richness and diversity indices associated with vaginal community state types (CSTs). (**a**) Sobs index of OTU level and (**b**) Shannon index of OTU level. * *p* < 0.05, ** *p* < 0.01, *** *p* < 0.0005.

**Figure 4 biology-11-01797-f004:**
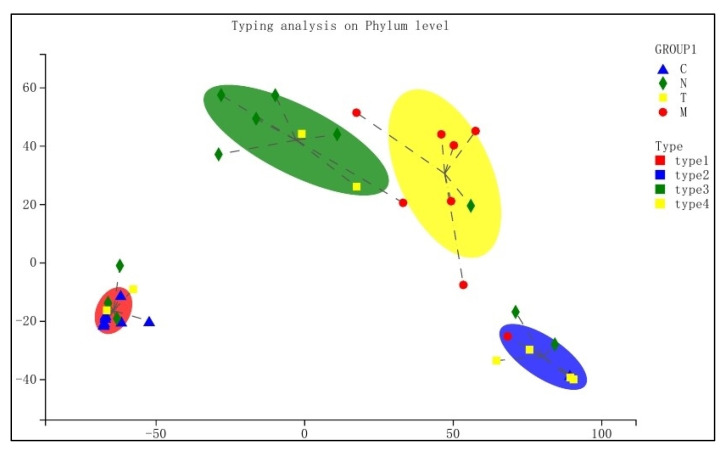
Results of the typing analysis at phylum level, with “groups” referring to different sample groups and “types” referring to different bacterial community composition types. Areas within circles indicate confidence intervals.

**Figure 5 biology-11-01797-f005:**
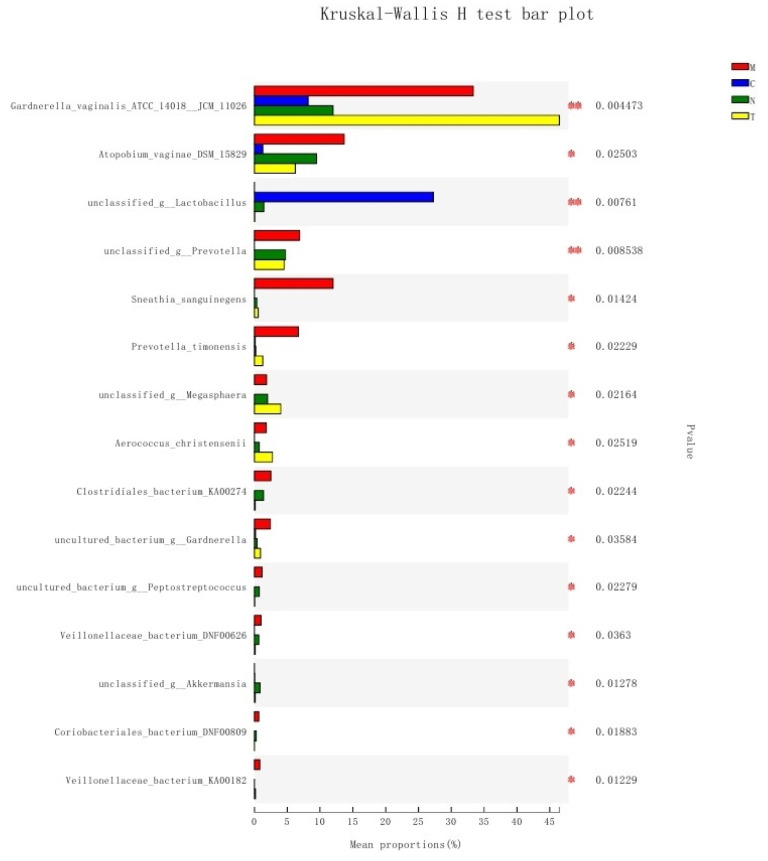
Distribution of bacterial communities in the four groups, with the average relative abundances of species in different groups plotted on the *x*-axis and species names plotted on the *y*-axis. Different colored columns indicate different groups. * 0.01 < *p* ≤ 0.05, ** *p* ≤ 0.01.

**Table 1 biology-11-01797-t001:** Summary of the available characteristics of patients in the four groups.

Characteristic		N (*n* = 12)	C (*n* = 11)	M (*n* = 9)	T (*n* = 8)	Total (*n* = 40)	*p*-Value
Age, years (mean ± SD)		37.50 ± 8.81	34.91 ± 6.24	42.67 ± 10.50	38.13 ± 8.29	38.10 ± 8.63	0.260
HPV status, n/N (%)	HPV (−)	12/12(100.00)	10/11(90.91)	7/9(77.78)	7/8(87.50)	36/40(90.00)	0.408
	HPV (+)	0/12(0.00)	1/11(9.09)	2/9(22.22)	1/8(12.50)	4/40(10.00)	
Pathological examination results	NILM	12/12(100.00)	8/11(72.72)	5/9(55.56)	5/8(62.50)	30/40(75.00)	0.271
	LSIL	0/12(0.00)	1/11(9.09)	3/9(33.33)	2/8(25.00)	6/40(15.00)	
	HSIL	0/12(0.00)	2/11(18.18)	1/9(11.11)	1/8(12.50)	4/40(10.00)	
Nugent score		0.58 ± 0.51	0.64 ± 0.67	7.56 ± 0.53	2.25 ± 3.06	2.50 ± 3.15	0.000
*Lactobacillus*		3.42 ± 0.51	3.55 ± 0.52	0.33 ± 0.50	3.00 ± 1.41	2.68 ± 1.49	0.000
*Gardnerella vaginalis* and *Bacteroides*		0.00 ± 0.00	0.18 ± 0.60	3.89 ± 0.33	1.25 ± 1.83	1.18 ± 1.77	0.000
*Mobiluncus*		0.00 ± 0.00	0.00 ± 0.00	0.00 ± 0.00	0.00 ± 0.00	0.00 ± 0.00	--

(1) Normal control group (N) consisting of healthy subjects (*n* = 12); (2) case–control group (C) consisting of subjects with clinical symptoms of BV and a Nugent score < 7 (*n* = 11); (3) disease group (M) consisting of subjects with clinical symptoms of BV and a Nugent score ≥7 (*n* = 9); (4) treated group (T) consisting of subjects of the disease group whose clinical symptoms of BV had been resolved after treatment with metronidazole or clindamycin (*n* = 8). High-risk HPV positive: infection with HPV16, HPV18, and 12 other types of HPV (HPV31, 33, 35, 39, 45, 51, 52, 56, 58, 59, 66, and 68). HPV positive: infection with HPV types other than high-risk HPV types. HSIL, high-grade squamous intra-epithelial lesion; LSIL, low-grade squamous intra-epithelial lesion; NILM, no intra-epithelial lesion or malignancy.

**Table 2 biology-11-01797-t002:** Sequencing results.

Domain	Kingdom	Phylum	Class	Order	Family	Genus	Species	OTUs *
1	1	25	51	137	244	535	869	1674

* OTUs: operational taxonomic units.

**Table 3 biology-11-01797-t003:** Analysis of alpha diversity indices.

Estimators	C-Mean	C-SD	M-Mean	M-SD	T-Mean	T-SD	N-Mean	N-SD	*p*-Value					
									C-M	C-T	C-N	M-T	M-N	N-T
Sobs	70.455	54.764	58.778	20.614	61.75	43.824	211.08	191.18	0.554	0.715	0.029 *	0.857	0.029 *	0.045 *
ACE	101.9	63.542	102.07	48.078	86.702	41.282	234.46	182.67	0.994	0.563	0.033 *	0.493	0.048 *	0.038 *
Chao	92.789	65.227	83.894	28.826	81.244	45.249	229.5	183.93	0.709	0.672	0.029 *	0.886	0.030 *	0.039 *
Shannon	0.410	0.282	1.827	0.702	0.836	0.845	2.264	1.877	0.000 **	0.135	0.004 **	0.018 *	0.517	0.060
Simpson	0.812	0.156	0.264	0.142	0.682	0.332	0.387	0.361	0.000 **	0.268	0.001 **	0.003 **	0.351	0.081
Coverage	0.999	0.001	0.999	0.000	0.999	0.000	0.999	0.000	0.155	0.069	0.714	0.470	0.416	0.259

* *p* < 0.05, ** *p* < 0.005. Indicators of community richness included the Sobs, ACE, and Chao indices; indicators of community diversity included the Shannon and Simpson indices.

## Data Availability

The datasets used and/or analyzed during the current study are available from the corresponding author upon reasonable request.

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
