# Peer review of "High-Throughput Sequencing-Based Analysis of Changes in the Vaginal Microbiome during the Disease Course of Patients with Bacterial Vaginosis: A Case–Control Study"

_biology, 2022, doi:10.3390/biology11121797_

Round 1

Reviewer 1 Report

In the manuscript, the authors present the changes of vaginal microbiota during the disease course of patients with bacterial vaginosis such as healthy group, the onset (Nugent score <7), progression (Nugent score >7), and remission after antibiotic treatment. This review represents an interesting opportunity for the research field regarding bacterial vaginosis. However, some issues need to be addressed before processing further. My other comments are as follows:

1. The authors present only the sequencing results among four groups in the manuscript. Please clarify the meaning of the changes of vaginal microbiota in the result sections. Especially, it would be helpful to add the specific changes of vaginal bacteria and scientific meaning to the experimental results of in the composition and alpha diversity of the vaginal microbiota in the abstract.

2. In treated group (T), the authors selected the subjects who is treated with metronidazole or clindamycin. Please mention in the manuscript why the authors selected metronidazole or clindamycin-treated groups.

3. Since metronidazole and clindamycin have completely different mechanisms of drug action, their effects on bacteria are expected to be different, finally influencing to the vaginal microbiome. Are there some changes in vaginal microbiome between metronidazole and clindamycin-treated groups?

4. It is mentioned that case-control group (C) targeted patients with a Nugent score of 7 or lower. As shown in Table 1, the average of Nugent score is about 0.64, which doesn’t seem to be much different from healthy subjects (N). Please address this.

5. In Figure 5, the mean proportion of Gardnerella vaginalis seems to be increased in T group. Why is the mean proportion of Gardnerella vaginalis augmented after the treatment?

Author Response

In the manuscript, the authors present the changes of vaginal microbiota during the disease course of patients with bacterial vaginosis such as healthy group, the onset (Nugent score <7), progression (Nugent score >7), and remission after antibiotic treatment. This review represents an interesting opportunity for the research field regarding bacterial vaginosis. However, some issues need to be addressed before processing further. My other comments are as follows:

  1. The authors present only the sequencing results among four groups in the manuscript. Please clarify the meaning of the changes of vaginal microbiota in the result sections. Especially, it would be helpful to add the specific changes of vaginal bacteria and scientific meaning to the experimental results of in the composition and alpha diversity of the vaginal microbiota in the abstract.

Response: Thank you for pointing this out. We have amended and included the information requested in “Results” of the “Abstract” (Lines 32-37, Lines 44-47). Most of the changes related to the vaginal microbiota and their significance were also highlighted in the “Discussion” section.

  1. In treated group (T), the authors selected the subjects who is treated with metronidazole or clindamycin. Please mention in the manuscript why the authors selected metronidazole or clindamycin-treated groups.

Response: The Sexually Transmitted Infections (STI) Treatment Guidelines released by the United States Centers for Disease Control and Prevention (U.S. CDC; July 2021), the Guidelines for Diagnosis and Treatment of Bacterial Vaginosis (2021 revised edition), and the Infectious Diseases Collaborative Group of the Society of Obstetrics and Gynecology of the Chinese Medical Association recommend clinicians to use metronidazole and clindamycin for the treatment of BV. This has been included in the manuscript at lines 71–76. Based on these guidelines, we selected metronidazole and clindamycin for the treatment of BV.

  1. Since metronidazole and clindamycin have completely different mechanisms of drug action, their effects on bacteria are expected to be different, finally influencing to the vaginal microbiome. Are there some changes in vaginal microbiome between metronidazole and clindamycin-treated groups?

Response: Both metronidazole and clindamycin have anti-anaerobic effects and are recommended treatment options in the “Guidelines for Diagnosis and Treatment of Bacterial Vaginosis (2021 revised edition) and the Infectious Diseases Collaborative Group of the Society of Obstetrics and Gynecology of the Chinese Medical Association”. Thus, patients whose symptoms resolved after treatment with these two antibiotics, did not experience relapse within 6 months, and were analyzed as a single group in the study.

  1. It is mentioned that case-control group (C) targeted patients with a Nugent score of 7 or lower. As shown in Table 1, the average of Nugent score is about 0.64, which doesn’t seem to be much different from healthy subjects (N). Please address this.

Response: The case-control group (C) refers to patients with clinical symptoms of BV with a Nugent score under 7. The healthy subjects (N) group refers to those who underwent routine physical examination at our hospital and had no history of BV or clinical symptoms of infection within 6 months, received normal results on routine examination of vaginal discharge with no Trichomonas, Candida, or clue cells detected, and received no antibiotic treatment within 2 weeks. The Nugent score of these individuals should be under 7. The main difference between groups C and N is whether clinical symptoms associated with BV were present, and the Nugent scores are indeed consistent between the groups.

  1. In Figure 5, the mean proportion of Gardnerella vaginalis seems to be increased in T group. Why is the mean proportion of Gardnerella vaginalis augmented after the treatment?

Response: Metronidazole and clindamycin are antibacterial drugs with good bactericidal activity against anaerobic bacteria, which were more numerous in the vagina and represented a higher proportion of the microbiome. When anaerobic bacteria were killed, the absolute number of Gardnerella vaginalis decreased substantially, but the relative number may have increased compared with that before treatment.

Reviewer 2 Report

It is an interesting and novel approach to vaginal microbiome

Some comments from my side

Methods

There was a sample calculation? Why 40 women was the selected number of participants

Is not clear how you define healthy subjects?  without symptoms? without discharge? never treated?

It will useful to describe briefly Nuget score

Results

Could influence HSIL status in the microbiome when is compared as unique variable among groups (there will be an extra damage in the tissue).

In normal patient which was the  status of chlamydia and Neisseria do you detect this germs on the other groups?

Author Response

1.Methods:There was a sample calculation? Why 40 women was the selected number of participants. Is not clear how you define healthy subjects?  without symptoms? without discharge? never treated? It will useful to describe briefly Nugent score

Response: Initially the number of the participants was more than 40. However, this point was mentioned in the "limitations of the study" in the final paragraph in the "Discussion" section. only a small number of patients were included in the various groups of the study due to the difficulties incurred in case matching and loss in follow-up after treatment (Lines 399-400). Sample collection was detailed in the subsection “2.1 Study population”. Healthy controls refer to those who underwent routine physical examination at our hospital during the same period as the study and had no history of BV or clinical symptoms of infection within 6 months, received normal results on routine examination of vaginal discharge with no Trichomonas, Candida, or clue cells detected, and received no antibiotic treatment within 2 weeks (Lines 115–119). The Nugent score is described in lines 57 to 60 of the “Introduction” section.

2.Results: Could influence HSIL status in the microbiome when is compared as unique variable among groups (there will be an extra damage in the tissue).

In normal patient which was the status of chlamydia and Neisseria do you detect this germs on the other groups?

Response: Our previous studies have revealed that the vaginal microbiota is directly or indirectly related to the progression of squamous intra-epithelial neoplasia, and Delftia may be a microbiological hallmark of cervical pre-cancerous lesions; these findings were published in 2020. Delftia was not detected in the present study, and the differences in the pathological examination results among the four groups were not statistically significant, so the relationship of HSIL status with vaginal microbiota in each group was not discussed in depth in the present study. Additionally, the colony abundance of Chlamydia in all four groups was very low in the present study, and the difference was not statistically significant; the organism might have been in a colonizing state. In addition, Neisseria was not detected.